# Predicting non-visible future tumour from baseline low dose CT using deep learned features

**Sheng Yu**[1]                                                   SHENG.YU@ICR.AC.UK
[1] *Institute of Cancer Research, UK*

**Yan Wen**[2]                                                    YWEN@LINCOLN.AC.UK
[2] *University of Lincoln, UK*

**Carolyn Horst**[3]                                             CAROLYN.HORST@KCL.AC.UK
[3] *King's College, London, UK*

**Balaji Ganeshan**[4]                                           B.GANESHAN@UCL.AC.UK
[4] *University College, London, UK*

**Spencer Thomas**[5]                                            SPENCER.THOMAS@NPL.CO.UK
[5] *National Physical Laboratory, UK*

**Reyer Zwigelaar**[6]                                           RRZ@ABER.AC.UK
[6] *Aberystwyth University, UK*

**Richard Lee**[7]                                               RICHARD.LEE@RMH.NHS.UK
[7] *The Royal Marsden NHS Foundation Trust, UK*

**Xujiong Ye**[2]                                                XYE@LINCOLN.AC.UK
**Matthew D. Blackledge**[1]                                     MATTHEW.BLACKLEDGE@ICR.AC.UK

**Editors:** Under Review for MIDL 2024

## Abstract

Studies have shown that yearly screening with low-dose computed tomography (LDCT) effectively reduces lung cancer mortality (NLST Research Team, 2011b). With the increasing number of deep learning tools that are trained on large collection of scans it has been possible to automatically report lung nodules (Venkadesh et al., 2023). We hypothesized that deep learning might also be able to predict the risk of future malignancies based on LDCT imaging in which no tumours are presently visible. This would provide a triage mechanism for identifying patients who would benefit from yearly screening versus those who might attend biannual screening. We use data from The National Lung Screen Trial (NLST) (NLST Research Team, 2011a) and compare the accuracy of multiple pre-trained classification models from the Keras library to predict a slice-by-slice risk of future tumour occurrence. The best performing model can achieved an AUC of 0.741 demonstrating a successful classifier.

**Keywords:** Lung cancer screening, Tumour prediction, Image registration.

## 1. Introduction

Lung cancer, with 50,000 new UK cases yearly, has a 10% ten-year survival rate, dropping to 5% for stage IV (Cancer Research UK). Annual low-dose computed tomography (LDCT) screenings enable early diagnosis and significantly improving survival chances by detecting the disease when it's most treatable (Toumazis et al., 2023). However, incidence of disease detection through LDCT screening is low (Adams et al., 2023). We explore the use of deep learning for risk-based triage to discern who needs annual versus biannual screenings, aiming to optimise healthcare resources efficiently.

## 2. Methods and datasets

### 2.1. Data Preparation

The NLST trial dataset was filtered to include patients on the LDCT arm for whom cancer was diagnosed at baseline (year 0), or at the first or second follow-up LDCT examinations (year 1 and year 2). Patient numbers in each of these categories were 240 for year 0, and 129 and 153 for years 1 and 2 respectively. All LDCT images were resampled to the same base pixel resolution using linear interpolation. A certified radiologist with expertise in thoracic imaging manually delineated all regions of interest (ROIs) around disease identified on the year 0, 1 and 2 scans. To determine the location of future cancers that occur within a year, a non-linear registration pipeline was developed to register LDCT imaging studies (and associated ROIs) from year 1 to year 0, and from year 2 to year 1, as demonstrated Figure 1)

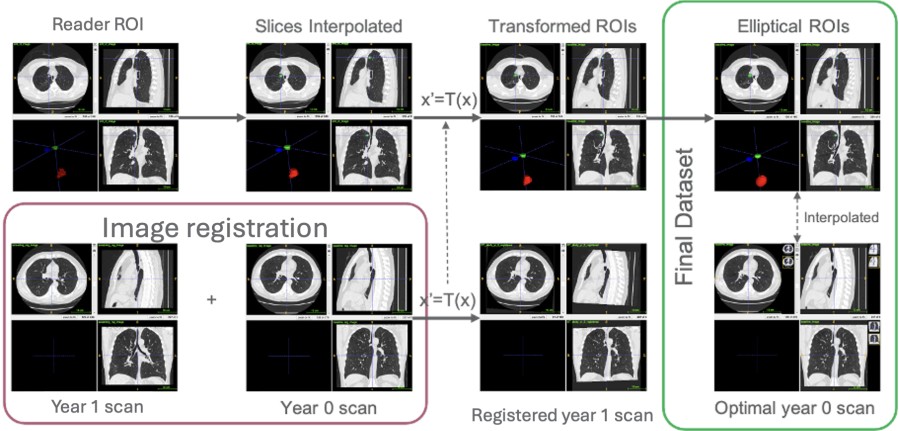

Figure 1: Our image registration pipeline was initialized by first segmenting the thoracic cavity using a publicly available lung mask model (Hofmanninger et al., 2020) and aligning the two resultant masks using affine transformation. It was further refined by B-spline (non-rigid) registration. The transforms were finally applied to the radiologist ROIs so that their location could be found on the year 0 scans.

### 2.2. Feature extraction

A total of 71 pre-trained deep learning models were used to extract deep features of the CT images. The weights of pre-trained models were obtained from the Keras project (Chollet et al., 2015). The complete list is labeled in Figure 2a. The input to the deep learning models is a 3D array that consists of a slice of the LDCT and its immediate neighbouring two slices. All arrays were preprocessed and normalised to comply with the requirement of each deep model architecture (Chollet et al., 2015). A feature vector (range from 368 to 4032 features) was obtained for each 3D slice array and they were used to train a classifier to identify which slices of the LDCT contain future tumours.

## 2.3. Model training and permutation test

522 scans (83659 slides) were split into a train (80%) and validation set (20%), where equal proportions of patients with different diagnostic years were maintained. A Gradient Boosting Tree Model (Ke et al., 2017) was trained for each feature sets that were generated by a deep learning model to predict whether a slice would contain a future tumour. The positive classes are the slices that contain registered tumour ROIs, whereas the rest of the slices are considered negative. After training, the performance of classifiers were evaluated in the independent validation set and we report the area under the receiver operating characteristic curve (AUC). Permutation tests (Ojala and Garriga, 2010) were used to estimate classifier performance significance by establishing a null hypothesis through label shuffling.

## 3. Result and conclusion

Figure 2a shows the performance of the classifiers ordered by their AUC metric as blue and their corresponding distribution generated by the permutation tests as box plots. The true model performances are significantly better than the null distribution of model with permuted labels. Figure 2b are examples of individual slice predictions from the test set. The x axis represents slide number and y axis is the probability output from the model using Youden's index (Youden, 1950) as a cutoff threshold.

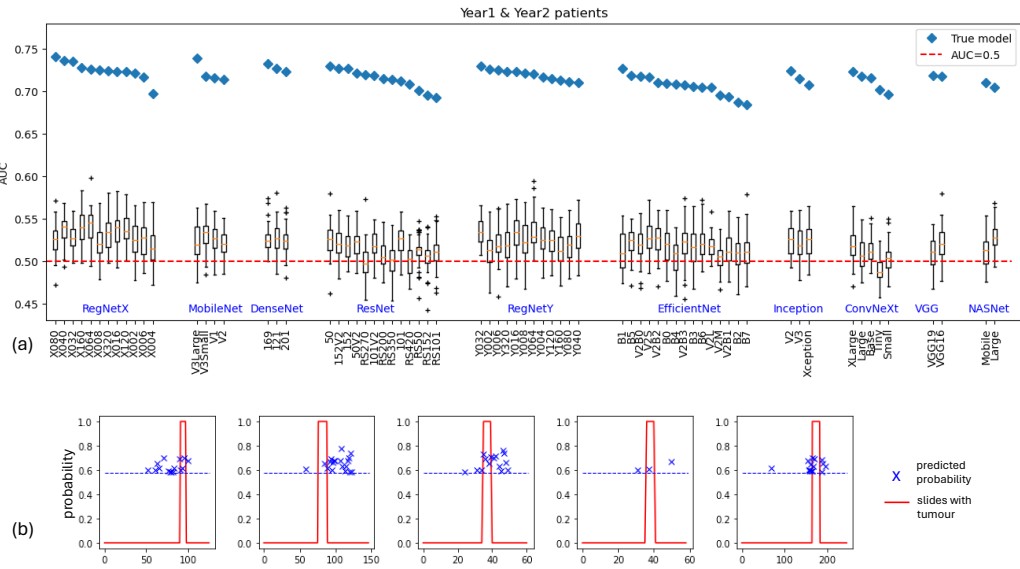

Figure 2: Boxplot of AUC ROC of trained classifier to identify future tumour slices

The permutation tests have demonstrated a moderate ability of features extracted from pre-trained deep models to predict locations of future tumour regions within one year from apparently non-visible baseline scans. This could help optimise lung screen frequency. Future work is needed to verify these results on larger datasets, and also include patients who do not have cancer diagnosis at years 0, 1 or 2.

## Acknowledgments

We thank Cancer Research UK for funding this work

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
