# OpenReview forum: "Predicting non-visible future tumour from baseline low dose CT using deep learned features"
_MIDL.io/2024/Short_Papers — MIDL 2024 Short Papers_

### Official Review · Reviewer_ZNpj · 2024-04-24

**Confidence:** 3
**Final Rating:** 3.5

**Review:**

This paper presents a method to predict whether a certain region of a low-dose chest CT image will develop into lung cancer in the future. For this, authors use data from the NLST trial with multiple time points. In the region where the cancer is known to be found in future scans, they crop a region and take three adjacent slides, and train a classifier with a broad set of features to predict the future presence of cancer.

#### PROS
* This is an interesting idea and a relevant question in the context of lung cancer screening
* Preliminary results are somehow promising

#### CONS
* It is unclear how three consecutive CT slices can be directly applicable to models pre-trained on RGB images; if some form of normalisation or rescaling is done, it's not described in the paper; if others have done the same in the same way as presented here, it would have been useful to add a reference.
* It is unclear why multiple backbones are used as feature extractors, why not only one? And if many are needed, why those ones? Another approach would have been to take a model pre-trained on CT and fine-tune it for the task at hand.
* It is unclear why scans at baseline were annotated, as the goal is to predict cancer within one year; I see that tutors present at baseline are segmented by an expert, probably needed for registration?
* Are axial slices resampled before feature extraction? If yes, was the input size the same for all networks used, or for some, slices had to be resampled in different ways? Could this affect the way some features were encoded?

---

### Decision · Program_Chairs · 2024-04-26

Accept